# Study of Particle Properties of Different Steels Sprayed by Arc Spray Process

**Rodolpho F. Vaz** [1,*] **, Anderson G. M. Pukasiewicz** [2]**, Hipólito D. C. Fals** [2] **,**
**Luciano A. Lourençato** [2] **and Ramon S. C. Paredes** [3]

[1] Thermal Spray Center CPT, University of Barcelona, Carrer Martí i Franques 1, 7a planta, 08028 Barcelona, Spain

[2] Mechanical Department, Federal University of Technology Paraná, Av. Monteiro Lobato, km04, 84016-210 Ponta Grossa-PR, Brazil; anderson@utfpr.edu.br (A.G.M.P.); hipolitofals@utfpr.edu.br (H.D.C.F.); lalouren@utfpr.edu.br (L.A.L.)

[3] Mechanical Department, Federal University of Paraná, Av. Coronel Francisco H. Santos, 210, 81531-980 Curitiba-PR, Brazil; ramon@ufpr.br

\* Correspondence: rvaz@cptub.eu

**Abstract:** Thermally sprayed coatings are employed for many purposes, as corrosion protection, wear resistance improvement, resistance to high temperatures, and others. The coating performance depends on its morphology, which is composed by splats, pores, oxide inclusions, and entrapped unmelted or resolidified particles. In arc spray process (ASP), the heat source is the arc electric obtained from the contact of two consumable metallic wires with different electric potentials, and the carrier gas is the compressed air. The velocity, dimensions, and thermal characteristics of the droplets sprayed are related to the morphology and properties of the coating. The main goal of this research is to evaluate how the velocity, temperature, and particle size are modified by the chemical composition of different materials (carbon steel, stainless steels, and FeMnCrSiNi alloy). The intention is to predict how the modification of the process parameters will change the particles properties. The materials had similar behavior tendencies during the flight: the velocity increased to a peak value then decreased, but this maximum value was different for materials with different particle size. The particles' size did not present significant differences during the flight; and the particles cooled down as they moved away from the gun, except the austenitic stainless steel and the FeMnCrSiNi alloy, which increased the droplets temperature during the travel. These alloys also presented more variation in chemical composition during flight.

**Keywords:** thermal spray; particles properties; arc spray process; in-flight

## 1. Introduction

Thermally sprayed coatings have many applications nowadays. Some examples of applications of the application of metallic and non-metallic materials are mentioned in the literature, as ceramic thermal barrier coating on different parts of gas turbines and internal combustion engines improve their efficiency and performance [1,2]; Aluminum, stainless steels, zinc, and other alloys coatings increase the corrosion resistance and consequently the life of offshore piping [3–5]; MCrAlY alloys coatings improve the resistance to hot corrosion and oxidation of gas turbines blades [6,7]; WC, TiC, SiC, and others carbides improve the wear resistance [7,8], or cavitation erosion resistance of hydraulic turbines. For this last purpose, some authors have studied FeMnCrSiNi austenitic alloys [9–12], which have some characteristics and properties adequate to absorb the energy from the cavitation phenomena without mass loss, where these properties are the pseudo-elasticity and the strain induced phase transformation [13–15].

One of the various thermal spray processes is the Arc Spray Process (ASP) or Twin-Wire Electric Arc (TWEA), which consists of the fusion of metallic alloys due to arc electric caused by contact of two consumable metallic wires, and this molten material is carried by gas to a prepared surface [5,16], as shown in Figure 1a. This process has some advantages if compared to other thermal spray processes, like easy operation, mainly for manual applications and in field applications, low-cost equipment and high deposition rate [5,7]. In some cases, the ASP can substitute the welding process, due to its lower heat input to the base material or substrate. An example of this substitution is the recovering of thickness of piping eroded or corroded on heat transfers or boilers [8,9].

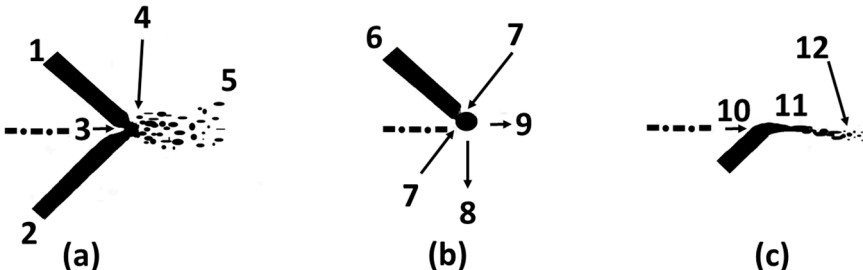

**Figure 1.** (**a**) Scheme of the arc spray process (ASP) principle. 1—positive wire; 2—negative wire; 3—atomizing gas direction; 4—molten material; 5—plume; (**b**) Forces acting during droplet formation on positive wire: 6—surface tension; 7—Pinch Effect; 8—droplet weight; 9—drag by the atomizing gas; (**c**) Formation of droplets on negative wire: 10—atomizing gas direction; 11—heat source or electric arc area; 12—droplets.

The plume or jet of droplets in ASP is due to the fusion of wire format feedstock. The process of origination of the droplets is different on positive and on negative wires. On the positive wire, the molten material is under the action of different forces, the weight, drag force by the carrier gas, particles surface tension, and electromagnetic force [16], indicated in Figure 1b. The resultant of these forces keeps the material attached to the wire until the pinch effect, caused by the increase of electromagnetic force, similar to what occurs to the wire on the Gas Metal Arc Welding (GMAW), this effect separates the droplet to the wire, but this effect happens on positive wire only in a frequency of 500 to 2800 Hz [17]. Elsewhere, the negative wire is warmed by the Joule effect and eroded by the carrier gas, originating the droplets, indicated in Figure 1c, like what happens in the Flame Spray (FS) process with wire feedstock [18]. From the different origin, the particles from the positive wire are more spherical and uniform, and the particles from negative wire tend to be more irregular and larger [18,19]. This heterogeneity of the particles' geometry produces different in-flight droplet behavior, from the gun to the substrate, bigger particles have more inertia to improve the velocity and to transfer heat to the environment, reaching the substrate hotter.

The ASP process parameters have a direct influence on the size of the droplets. The droplets are smaller, as is the voltage and carrier gas pressure higher with current and wire diameter increase [5,19]. The droplet size influences the coating morphology, bigger droplets form large lamellae or pancakes. This relation is important since the coating properties and performance depend on the lamellae properties and sizes [20,21].

Other influences on the coating properties are the characteristics of the particles during flight and the changes of these characteristics while traveling from the gun to the substrate. Environment also influences the droplets, causing variations in their size, temperature, and velocity to occur. Others authors monitored and measured these properties to study the properties of different coatings and thermal spray processes. Some examples are yttria-stabilized $ZrO_2$ by Atmospheric Plasma Spraying (APS) [22–24], NiCr by High Velocity Oxy-Fuel (HVOF) [25] and ASP [26], 316L by HVOF and APS [27], chrome steel by ASP [5,26], and aluminum by Flame Spraying (FS) [18], Cold Gas Spraying (CGS) [28], or ASP [29].

Chemical composition of the droplets can also change during flight, mainly due to the reaction of oxidation of the material, if compressed air is used as atomizing gas [30]. The particles are under oxidation since their formation in heat source or gun until the end of the solidification in the coating [31]. The increase of the particles velocity reduce their oxidation during the flight, as well as the pre-solidified particles, porosity, and oxide content in the coating [20].

The ASP is broadly used to deposit diverse alloys on diverse structures. Many works related the relation between parameters of deposition and characteristics of the coatings [3,7,8,26,31,32]; however, the behavior of the particles, during flight had not been much explored, concerning the influence of the chemical composition of the steel on the properties of the droplets.

This work explores the evolution of diameter, velocity, temperature, and chemical composition of particles sprayed by ASP. The materials evaluated were four different steels (70S-6, 309L, 410NiMo, and FeMnCrSiNi) and the measures were done at 40, 80, 120, 160, and 200 mm from the gun. The results of velocity, temperature, and size of particles are expressed in statistical normal and histograms, since the range of values achieved each experiment.

## 2. Materials and Methods

The thermal spray equipment used in these experiments was the Sulzer Metco Value Arc 300E with Sulzer Metco Arc Gun LCAG (Westbury, NY, USA). The parameters are shown in Table 1 and the sprayed materials were solid wires with diameter 1.20 mm with the chemical composition shown in Table 2. The exception is the special alloy, FeMnCrSiNi, which was flux-cored wire with diameter 1.60 mm produced by Durum do Brasil (Sorocaba, SP, Brazil).

**Table 1.** Parameters used for ASP.

| Voltage | Current | Standoff Distance | Atomizing Gas Pressure |
|---------|---------|-------------------|------------------------|
| 30 V | 180 A | 40 mm | 55 psi (380 kPa) |
| | | 80 mm | |
| | | 120 mm | |
| | | 160 mm | |
| | | 200 mm | |

**Table 2.** Predicted Thermally sprayed alloys chemical composition wt%.

| Material | C | Cr | Ni | Mo | Mn | Si | B | Fe |
|----------|---|-----|-----|-----|-----|-----|---|-----|
| AWS A5.18 ER70S-6 [33] | 0.06 0.15 | – | – | – | 1.40 1.85 | 0.80 1.15 | – | Bal. |
| AWS A5.9 ER309LSi [34] | 0.03 | 23.00 25.00 | 12.00 14.00 | 0.75 | 1.00 2.50 | 0.65 1.00 | – | Bal. |
| AWS A5.9 ER410NiMo [34] | 0.06 | 11.00 12.50 | 4.00 5.00 | 0.40 0.70 | 0.60 | 0.50 | – | Bal. |
| FeMnCrSiNi [35] | 0.50 max | 8.00 16.00 | 5.00 max | – | 10.00 30.00 | 2.00 6.00 | 2.00 max | Bal. |

To identify the geometry of the particles in flight, the plume of particles sprayed was collected on water and these particles were prepared and observed on Scanning Electron Microscopy (SEM), and we also analyzed the variation of chemical composition using technique of Energy-Dispersive Spectroscopy (EDS). In the solid wire (70S-6, 309L, and 410NiMo), areas of 1 mm$^2$ were mapped on cross section; for the flux-cored wire (FeMnCrSiNi), the wire was fused on a plasma jet of Gas Tungsten Arc Welding (GTAW) torch and on this solidified material an area of 1 mm$^2$ was mapped. Ten measured points were made in each condition. In Figure 2 are shown examples of the surface

chemical analysis by EDS. The SEM used in this research was a Tescan Vega 3 (Brno, Czech Republic), and the EDS equipment was an Oxford Instruments X-Max 50 with Aztec software (Abington, UK).

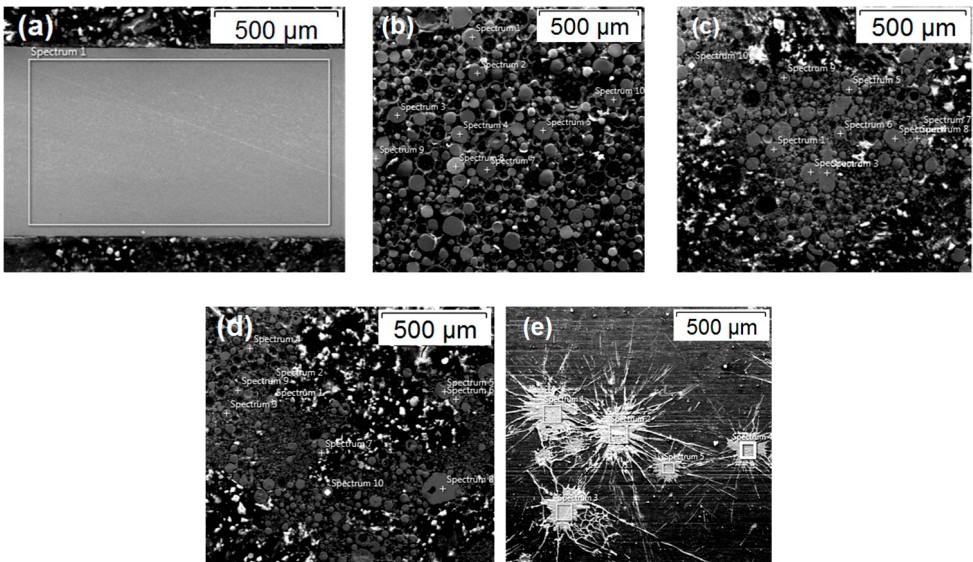

**Figure 2.** Images of the location where the chemical composition analyses were realized. (**a**) Wire of 309L, the rectangle Spectrum 1 delimits the area for the Energy-Dispersive Spectroscopy (EDS) measurement; (**b**) FeMnCrSiNi, (**c**) 410NiMo, and (**d**) 309L particles, the points spectrum indicate in which particles the EDS was done; (**e**) 70S-6 lamellae deposited in flat glass, the rectangles' spectrum delimit the area where the EDS was done in each splat.

To measure the particles properties, the equipment Tecnar DPV eVolution (Saint-Bruno-de-Montarville, QC, Canada) was used. This equipment had the setup to measure 5000 particles at different distances from the gun and create histograms containing the velocity, temperature, and the particle diameter. Figure 3 shows the equipment used during the depositions.

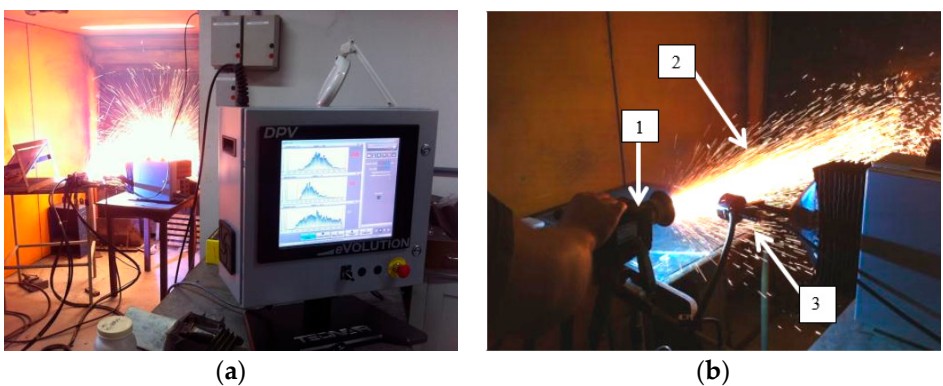

|(**a**)|(**b**)|

**Figure 3.** Tecnar DPV Evolution equipment assembly for the ASP process. (**a**) Monitor presenting results online; and (**b**) position of the sensor, with 1—ASP gun; 2—plume; and 3—DPV sensor.

## 3. Results and Discussions

### 3.1. Variation of Diameter of Droplets

The particles presented a tendency to decrease their size or diameter during flight. The alloy 309L had different behavior because of its chemical composition, mainly the content of Cr, which has high reactivity to the oxygen from the environment in high temperature, oxidizing and increasing the size of particles [36]. The FeMnCrSiNi alloy had larger particles than the others alloys because this wire

was 1.60 mm diameter, against 1.20 mm diameter of the other materials, and the wire size has a direct influence on the droplets sizes [20].

The materials did not present any relevant variation of the size of the particles during the flight, such as the alloy 70S-6 started the travel with 45 ± 18 μm, reducing to 39 ± 18 μm, about 10% of variation, and this range is inside of the standard deviation, and the same occurs to all materials. An explanation for the high standard deviation observed in the diameter values is the difference in diameter between the droplets formed on the cathode wire and the droplets from the anode wire, which are much larger [16,18,19]. Another explanation is that these droplets are formed in situ (in the gun) from wires, which changed drastically the format, different to what occurs on other thermal spray process, like HVOF, APS, or CGS, for example [7], where the droplets are originated from powder previously sieved and carefully prepared by atomization in inert gas or another controlled process to the application.

The interpretation of the variation of the diameter was on the mean value and standard deviation, but the data collected belong to a statistical distribution for all the materials and distances of measurement, as presented in Figure 4. It is possible to verify in Figure 4 a normal distribution for all materials and distances, with a tendency to skewed to right format of the histogram, with a tendency to skewed to right format of the histogram, except for 309L, which maintained similar particles distribution during flight. From this, the amount of smaller particles is higher for 70S-6 and 410NiMo than for 309L and FeMnCrSiNi. Another observation is the constant reduction of the larger particles' diameter, due to the fragmentation of the droplets during the flight by the air compressed carrier gas action. In Figure 4 are presented three of the five histograms collected for each material. The histograms at 80 and 160 mm presented the same tendency observed in the sequence 40, 120, and 200 mm. In the supplementary files are provided all the histograms of diameter—Supplementary Figure S1–S4.

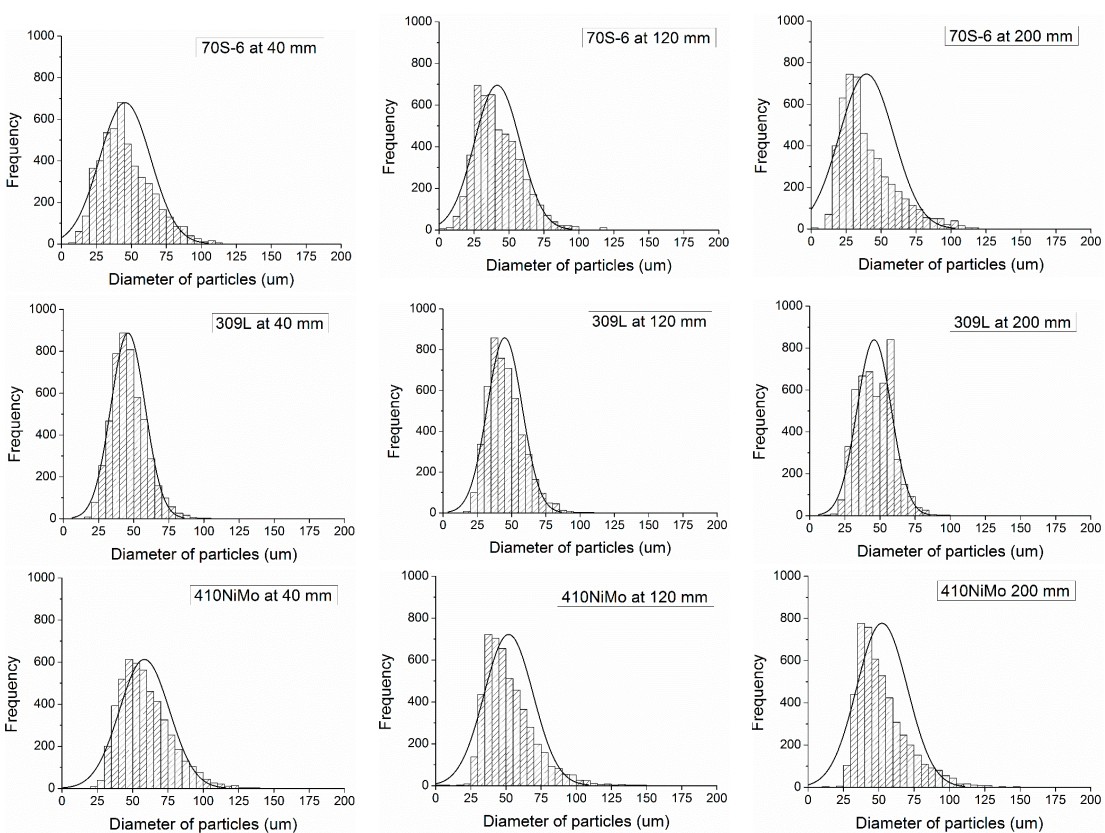

**Figure 4.** *Cont.*

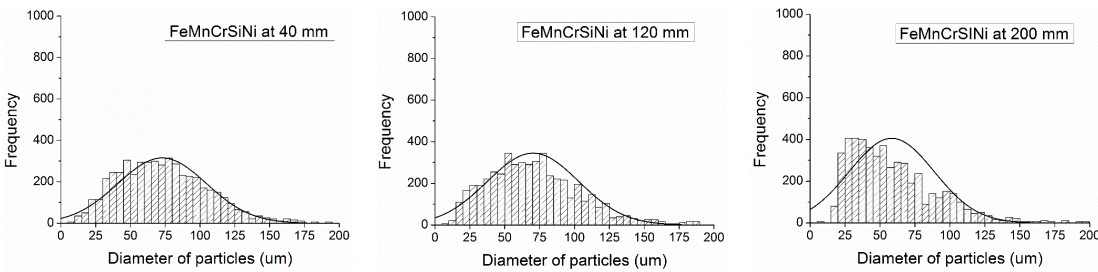

**Figure 4.** Histograms of diameter of particles observed in different distances from the gun.

One key point here is, beyond the diameter of the wire, the capacity to form a stable oxide formation in the stainless steel alloys and FeMnCrSiNi, because the addition of Cr and mainly Si form a more resistant oxide that reduces the droplets fragmentation by the viscosity increase of the particle by the oxide formation [37]. Meanwhile, the less oxidation resistance, 70S-6 carbon steel showed a formation of smaller particles, promoting a superficial area increase and consequently higher oxidation.

Therefore, the use of a lower oxidation resistance alloy must to be deposited with higher values of current, and lower values of air pressure, because of the higher atomizing process. The higher atomizing leads to a particle diameter reduction, increasing the oxidation, which can reduce the adhesion of the coating. On the other hand, wires with higher diameter and higher oxidation resistance material can be deposited with higher air pressure and lower current, which can promote a better adhesion.

SEM images of 410NiMo and 309L particles sprayed and captured in water at a distance of 200 mm from the gun are presented in Figure 5. From these images, it is possible to interpret the particles as spherical and relatively regular, similar to the format of particles obtained by other authors spraying in the same conditions and process [21], but with a huge range of sizes, as previously known from the histograms in Figure 4. Some isolated hollow particles of 309L and FeMnCrSiNi were found, as indicated by arrow in Figure 5b. This type of particle is typical of some powder alloys obtained by the atomization process [38,39], where the molten material is exposed to a jet of gas and is solidified very fast, very similar to what occur in the ASP process. Another aspect that must be observed is the oxide formation inside the particles because the rupture of the oxide layer during the flight [37].

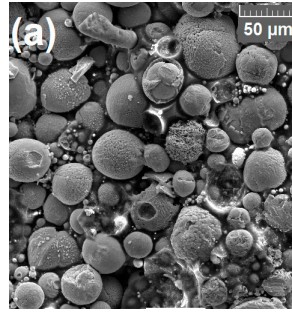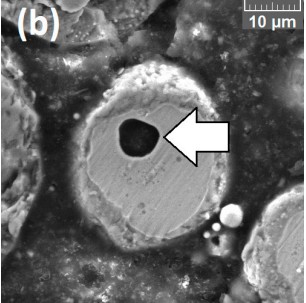

**Figure 5.** Scanning Electron Microscopy (SEM) of spherical geometry of particles collected in a distance of 200 mm from the gun. (**a**) 410NiMo with magnification of 500×, and (**b**) 309L with hollow, and oxide formation with magnification of 2500×.

### 3.2. Study of the Droplets Velocity

In the ASP process, the particles are molten in the contact of wires and dragged to the substrate by the atomizing gas, which was compressed air in this work. This gas is responsible to impose velocity to the droplets and this velocity increases from the gun to the peak value during the travel and then the velocity decreases, as seen by other authors spraying different materials [36,40,41]. This behavior was observed on the 70S-6 and 410NiMo materials, which had their maximum velocity at 160 mm from the gun: 126 ± 21 and 122 ± 17 m/s, respectively. These values are close to the highest value of 120 m/s obtained by others authors spraying steel by ASP with the same voltage (30 V) and air pressure

(0.4 MPa), but lower current (100 A) [5] However, the other alloys did not reach their maximum velocity and this velocity happens further than 200 mm from the gun.

The interpretation of the variation of velocity was on the average and standard deviation of the values, although, the data collected was in a statistical distribution for all the materials and distances of measurement, as presented in Figure 6, where is possible to see a normal distribution on the histogram at each distance of measure from the gun. It is possible to observe that the quantity of the particles with higher velocity increase with distance. The bigger and weightier droplets retard the increase of the velocity due their inertia and this effect is observed in the histograms of FeMnCrSiNi alloy at 40 and 120 mm, not forgetting that this alloy was fabricated in diameter 1.6 mm wire, while the all the others were diameter 1.2 mm wire. Thus, the increase on the wire diameter must be accompanied with an increase of the stand-off distance and air pressure. In Figure 6 are presented three of the five histograms collected for each material. The histograms at 80 and 160 mm presented the same tendency observed in the sequence 40, 120, and 200 mm. In the supplementary files are provided all the histograms of velocity—Supplementary Figure S5–S8.

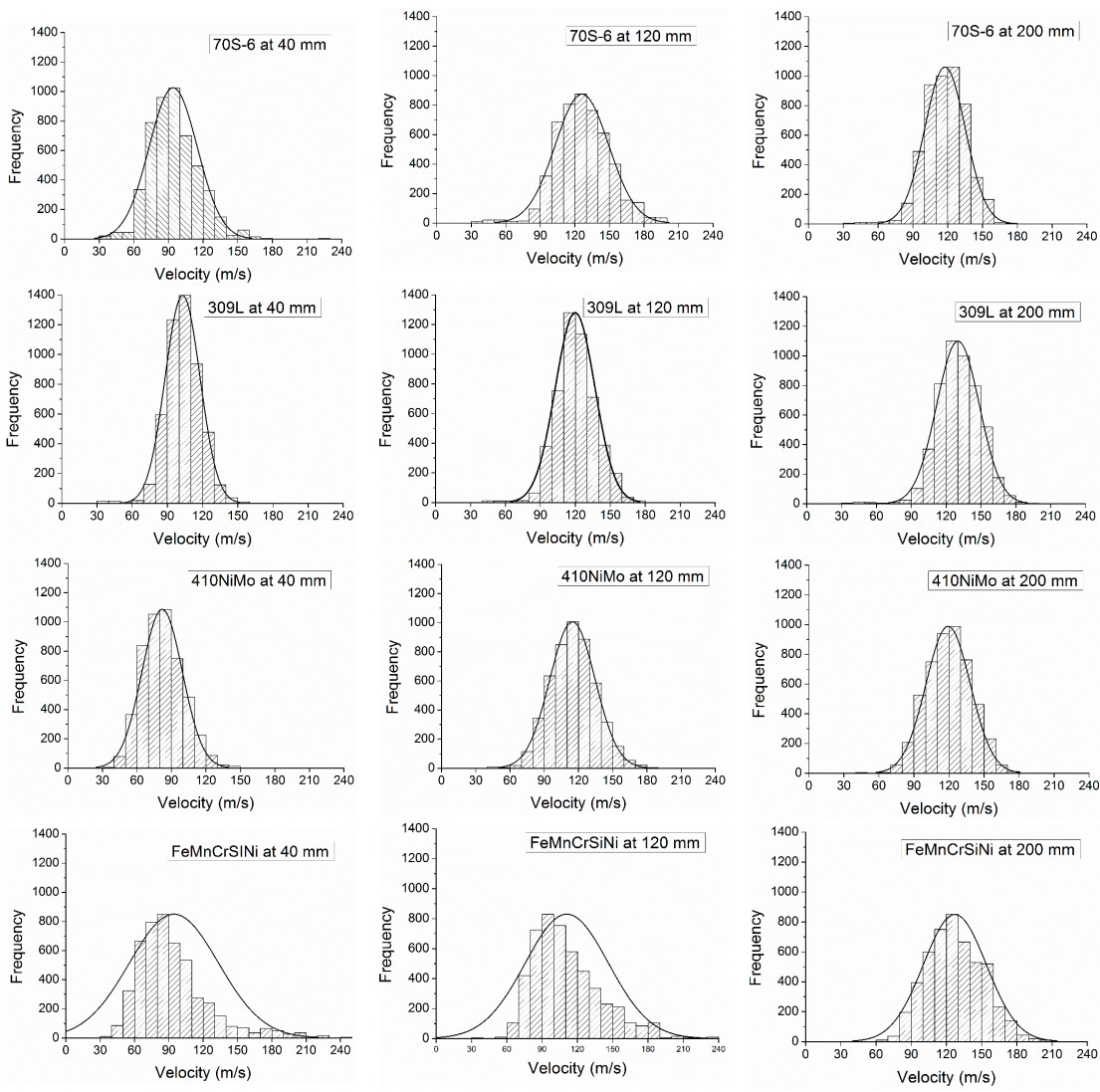

**Figure 6.** Histograms of the particles' velocity collected in different distance from the gun.

An important application for the knowledge about the distance where the particles have the peak velocity is to define the standoff distance to spray each alloy. For example, spraying the 70S-6 or 410NiMo alloys, the best standoff distance is 160 mm, where the Kinetic energy is the highest possible,

since mathematically this kind of energy has direct relation to the velocity, as seen in Equations (1), where K is the Kinetic energy, m is the particle mass, and V is the velocity of the particle. As higher is the energy of the droplets at the moment of impact, as higher is the adherence of the lamellae on the substrate surface and lower porosity [40–42].

$$K = \frac{m \cdot V^2}{2} \tag{1}$$

### 3.3. Study of the Droplets Temperature

During the flight, the droplets of 70S-6 and 410NiMo transferred heat to the environment, reducing their temperature. The droplets of 70S-6 had 2109 ± 105 °C at 40 mm and reduced progressively to 2024 ± 108 °C at 200 mm, near to 4% of variation; and the 410NiMo, presented reduction near to 2%. The average temperature of all alloys were close to each other, however, lower than the reference temperatures, 2150 ± 189 °C for the deposition of Ni by ASP [43] and 2120 °C for the spraying of steel by ASP [5].

However, the tendency of decrease the average temperature during the flight was not observed for the 309L and FeMnCrSiNi particles. The 309L increased the temperature from 1986 ± 97 °C at 40 mm to 1992 ± 78 °C at 200 mm, less than 1%, and the alloy FeMnCrSiNi started the travel with 1998 ± 157 °C at 40 mm and finished with 2027 ± 162 °C at 200 mm, variation of 1%. This effect is related to the alloys chemical composition, the content of Cr in 309L and of Cr, Mn, and Si in FeMnCrSiNi alloy is higher than in the 70S-6 or 410NiMo and these alloy elements oxidize easily is an exothermic chemical reaction [44], promoting the temperature increase during the flight. Another aspect that must to be observed is the diameter of the particles, bigger particles have less surface area in contact with the oxygen, reducing the oxidation and consequently the temperature of the particles. Therefore the lower diameter of the 70S-6 particles rise their oxidation and their temperature, reducing the viscosity which can collaborate with the particles diameter reduction.

The interpretation of the variation of velocity was on the average and standard deviation values, although, the data collected was in a statistical distribution for all materials and distances of measurement, as presented in Figure 7, where is possible to see a normal distribution and histograms, with a symmetrical format for all materials and distances, except for FeMnCrSiNi, which presented a tendency to symmetrical rectangular at 200 mm of distance, with a more broad distribution of the temperature. In the Figure 7 are presented three of the five histograms collected for each material. The histograms at 80 and 160 mm presented the same tendency observed in the sequence 40, 120, and 200 mm. In the supplementary files are provided all the histograms of temperature—Supplementary Figure S9–S12.

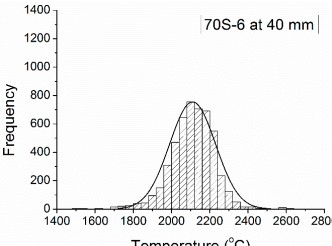 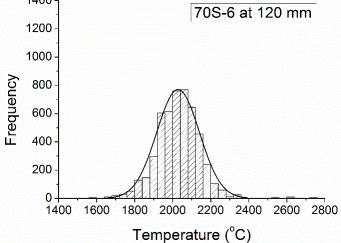 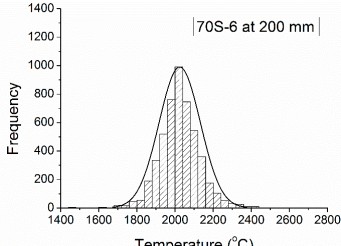

**Figure 7.** *Cont.*

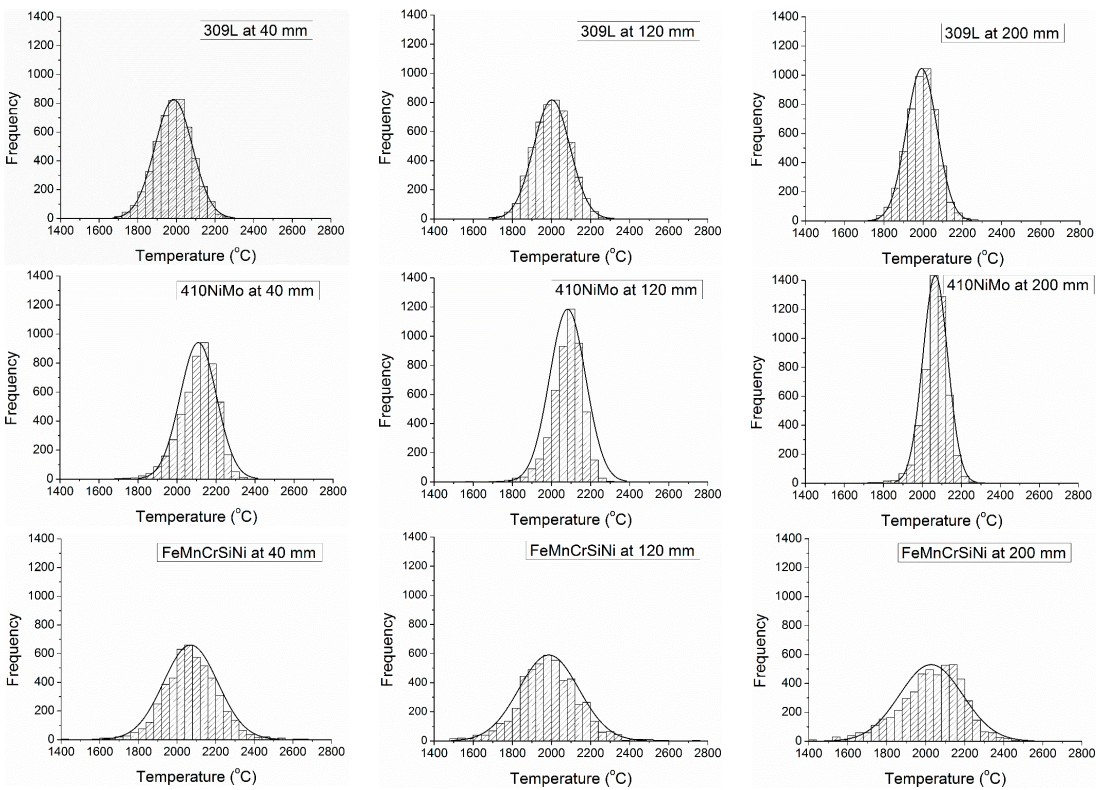

**Figure 7.** Histograms of temperature of particles versus distance from the gun.

### 3.4. Chemical Composition of the Droplets

The wire, particles, and lamellae chemical compositions were measured by the EDS technique for each sprayed material and the results are presented in Table 3. Elements with more affinity to oxygen presented reduction of their content during the flight of droplets from the gun, as seen in the content of Cr, Mn, and Si in all alloys. This behavior, including the variation of C, by formation of CO and $CO_2$, is common and was observed by other authors, spraying by ASP or other processes [21,26,30]. In Figure 2 are presented the conditions of the surfaces where the chemical compositions were measured. The oxidation of particles shall occur on the heat source, during the travel, and in the solidification of lamellae [30,31], nevertheless, for ASP, the first one is the most important, confirmed by the greater variation in chemical composition from the wire condition to the particle condition and less variation from the particle to lamellae conditions.

**Table 3.** Variation of chemical composition of sprayed materials (Mass %).

| Material | Condition | Cr | Ni | Mo | Mn | Si | Fe |
|---|---|---|---|---|---|---|---|
| 70S-6 | Wire | – | – | – | 1.5 ± 0.1 | 1.2 ± 0.1 | Bal. |
| | Particles | – | – | – | 1.2 ± 0.1 | 1.2 ± 0.1 | Bal. |
| | Lamellae | – | – | – | 1.2 ± 0.1 | 0.9 ± 0.1 | Bal. |
| 309L | Wire | 24.2 ± 0.2 | 13.0 ± 0.2 | 0.3 ± 0.2 | 2.0 ± 0.1 | 1.0 ± 0.1 | Bal. |
| | Particles | 21.4 ± 0.2 | 13.9 ± 0.2 | 0.4 ± 0.3 | 1.1 ± 0.1 | 0.2 ± 0.1 | Bal. |
| | Lamellae | 20.4 ± 0.2 | 13.2 ± 0.3 | 0.3 ± 0.2 | 0.9 ± 0.1 | 0.2 ± 0.1 | Bal. |
| 410NiMo | Wire | 12.9 ± 0.1 | 3.8 ± 0.1 | 0.5 ± 0.2 | 0.9 ± 0.1 | 1.2 ± 0.1 | Bal. |
| | Particles | 12.5 ± 0.1 | 4.1 ± 0.2 | 0.4 ± 0.3 | 0.7 ± 0.1 | 0.5 ± 0.1 | Bal. |
| | Lamellae | 12.4 ± 0.2 | 3.8 ± 0.2 | 0.4 ± 0.3 | 0.6 ± 0.1 | 0.3 ± 0.1 | Bal. |
| FeMnCrSiNi | Wire | 17.0 ± 0.6 | 5.0 ± 0.2 | – | 10.2±0.4 | 8.9 ± 0.3 | Bal. |
| | Particles | 17.3 ± 0.2 | 5.2 ± 0.2 | – | 7.1±0.3 | 6.3 ± 0.1 | Bal. |
| | Lamellae | 17.2 ± 0.6 | 5.2 ± 0.2 | – | 6.1±0.3 | 6.8 ± 0.3 | Bal. |

From the Ellingham diagram [44] is predictable the priority for oxidation of the alloy elements Si, Mn, Cr, Fe, and Ni. However, the small content of Si makes its variation imperceptible, except to the FeMnCrSiNi alloy, in which Si and Mn presented a significant reduction from the condition of wire to lamellae. The Cr presented a reduction of 17% on the 309L steel, but no sensitive value variation on others alloys. Part of this Cr reduction on the 309L is seen by the formation of oxides and their inclusion inside the droplets, as seen in the transversal sections of a polished particle in an SEM image in Figure 8a, compared to a particle of 70S-6 without internal oxidation, Figure 8b. The oxides are imprisoned in the particles due to its formation during the melting in the gun and not during flight, proving in the feedstock steel what was seen by other authors spraying aluminum by ASP [31].

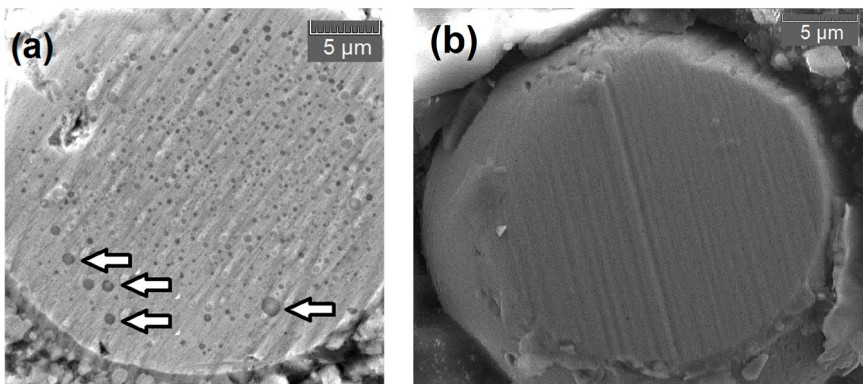

**Figure 8.** (**a**) imprisoned oxides in transversal section droplet of 309L. The arrows indicate few of many circular oxides. (**b**) Cross section of 70S-6 droplet do not have internal oxides.

## 4. Conclusions

Only severe modifications in the steel chemical composition produce significant variations on the characteristics of particles or droplets sprayed by ASP during the flight from the gun to the substrate. The velocity, temperature, and size are dynamics and changed with the same tendency to similar steel compositions and the alloys with more content of alloy elements, 309L and FeMnCrSiNi, had different behavior than 70S-6 and 410NiMo.

The particles formed in the heat source of the ASP process have a high degree of sphericity and regular particle size distribution, with a large range and standard deviation. The smallest particles were found in the 70S-6 and 410NiMo wires and reached maximum speed before the largest and heaviest particles, found in the 309L and FeMnCrSiNi wires. The largest particles were observed on the wire with the largest diameter and oxidation resistance material (FeMnCrSiNi).

The presence in the wires of reactive chemical elements, such as Cr, Mn, and Si, cause a small increase in temperature during the flight of the particles in the APS process; however, it has been verified that in materials with a lower content of these elements, the particles cool down during the flight. The chemical composition of the alloy sprayed varies from the wire condition (feedstock material) to the lamellae condition (condition of coating), reducing the content of oxidizing alloy elements, mainly Mn and Si.

The study of the behavior of the velocity and temperature of the particles is an important way to improve the properties of the coating, and as observed in this study, the chemical composition of the material changes the optimal deposition parameters. The improvement of the oxidation resistance of the material can lead to an increase in the distance of the deposition, and the opposite way, for lower oxidation resistance material. The deposition using wires with larger diameters must be optimized, using higher air pressure and standoff distance.

**Supplementary Materials:** The following are available online at http://www.mdpi.com/2079-6412/10/4/417/s1, Figure S1: Histograms of diameter of 70S-6 particles observed in different distances from the gun, Figure S2: Histograms of diameter of 309L particles observed in different distances from the gun, Figure S3: Histograms of diameter of 410NiMo particles observed in different distances from the gun, Figure S4: Histograms of diameter of FeMnCrSiNi particles observed in different distances from the gun, Figure S5: Histograms of velocity of 70S-6 particles observed in different distances from the gun, Figure S6: Histograms of velocity of 309L particles observed in different distances from the gun, Figure S7: Histograms of velocity of 410NiMo particles observed in different distances from the gun, Figure 8: Histograms of velocity of FeMnCrSiNi particles observed in different distances from the gun, Figure S9: Histograms of temperature of 70S-6 particles observed in different distances from the gun, Figure S10: Histograms of temperature of 309L particles observed in different distances from the gun, Figure S11: Histograms of temperature of 410NiMo particles observed in different distances from the gun, and Figure S12: Histograms of temperature of FeMnCrSiNi particles observed in different distances from the gun.

**Author Contributions:** Conceptualization, R.F.V. and A.G.M.P.; methodology, R.F.V. and A.G.M.P.; investigation, R.F.V.; resources, R.F.V., A.G.M.P. and R.S.C.P.; writing—original draft preparation, R.F.V.; writing—review and editing, R.F.V., A.G.M.P., H.D.C.F. and L.A.L.; supervision, R.S.C.P. All authors have read and agreed to the published version of the manuscript.

**Funding:** This research was partially funded by Copel Geração e Transmissão S.A., via R&D project registered in Brazilian National Agency of Energy ANEEL under code PD-6491-0023-2010 and executed by Institute of Technology for Development Lactec.

**Acknowledgments:** The authors wish to thank Federal University of Paraná UFPR, Post-Graduation Program in Mechanical Engineering of Federal University of Paraná PGMEC, and Federal University of Technology Paraná UTFPR, for the opportunity to develop the scientific experiments in their facilities; and to The Brazilian National Council for Scientific and Technological Development CNPq, for the tax incentive to import equipment to Brazil. Rodolpho F. Vaz also would like to thanks specially the colleagues Dr. Gustavo B. Sucharski and MSc. Eduardo A. Alberti, for the assistances during the experiments and depositions.

**Conflicts of Interest:** The authors declare no conflicts of interest. The funders had no role in the design of the study; in the collection, analyses, or interpretation of data; in the writing of the manuscript, or in the decision to publish the results.

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
