# Peer review of "Study of Particle Properties of Different Steels Sprayed by Arc Spray Process"

_coatings, doi:10.3390/coatings10040417_

Round 1
Reviewer 1 Report
This manuscript is rather a report than a scientific paper. I can not see any valuable point that deserves publication.
Author Response
Thanks to the reviewer, the criticism of the reviewer motivate the authors to improve the text. The authors believe that the importance of this paper is on the study of how the chemical composition of the material can affect the deposition conditions using ASP process. This paper can contribute to the scientific community and industry, since the materials evaluated (steels) are commonly applied and the results obtained can help the users of ASP process to confirm, or change the parameters already in use, mainly standoff distance, or arc tension to improve the quality of the coatings. Also, the alloy FeMnCrNi is new, under patent, and this evaluation of particles in-flight was never done before, configuring a novelty. The knowledge of their properties and behavior will support future studies of this alloy. The main goal of this research is evaluate how the velocity, temperature, and particle size are modified by the chemical composition of different materials (carbon steel, stainless steels, and FeMnCrSiNi alloy). The intention is predict how the modification of the process parameters will change the particles properties. The authors modified the text with intend to improve the writing, giving a more accurate view of the importance of this paper.
Reviewer 2 Report
In this article by Vaz et al. the authors prepared coatings by the arc spray process. The results are interesting and potentially valuable to wider audience, but several issues must be addressed first to warrant publication of this material. Please find the suggestions below:
1) I suggest expanding the affiliations, which currently do not specify where you are based. The authors may guess that two out of three authors represent Brazil based only on the email addressed provided.
2) Abstract: "The morphology of coating is composed by lamellae, porous or voids, and oxides" - this sentence is very confusing
3) The draft should be proofread to remove the errors. For instance "then" not "than" (Line 18), "improve" not "improving" (Line 29), "warmed" not "warm" (Line 48), etc.
4) The novelty factor is not clearly defined. The paragraph summarizing what you have done is just three lines long and does not link to the state of the art. Please include exactly what is new as compared to what has been done already.
5) Table 2 - these numbers do not add up to a 100%. What is the rest?
6) Fig. 4, 7 and 8 - professional scale bar markers should be provided
7) SEM micrographs are provided for a subset of experiments. Could you please enclose justification why exactly these samples were visualized?
8) Could you please comment on the reproducibility of this method? How many experiments were conducted for each parameter set?
Author Response
1) I suggest expanding the affiliations, which currently do not specify where you are based. The authors may guess that two out of three authors represent Brazil based only on the email addressed provided.
- Thanks to the reviewer, the authors improve the text, including the authors information. Two authors were included due the helping on the writing and discussion of the paper.
2) Abstract: "The morphology of coating is composed by lamellae, porous or voids, and oxides" - this sentence is very confusing.
- Rewrote: “The thermally sprayed coatings typically feature splats, pores, oxide inclusions, and entrapped unmelted or resolidified particles”.
3) The draft should be proofread to remove the errors. For instance "then" not "than" (Line 18), "improve" not "improving" (Line 29), "warmed" not "warm" (Line 48), etc.
- Thanks, the authors made an extensive revision, and we think that the text was improved.
4) The novelty factor is not clearly defined. The paragraph summarizing what you have done is just three lines long and does not link to the state of the art. Please include exactly what is new as compared to what has been done already.
- Reviewed and extended the text.
- This study have a contribution to the scientific community and to the industry, since the materials evaluated (steels) are commonly applied and the results obtained help the users of ASP to confirm the parameter already in use, mainly the standoff distance, or change to improve the quality of the coatings.
- The alloy FeMnCrNi is new, under patent, and this evaluation of particles in-flight was never done before, configuring a novelty. The understand of its particles characteristics and behavior will support future studies of this alloy.
5) Table 2 - these numbers do not add up to a 100%. What is the rest?
- Table edited. Inserted the Fe alloy element.
6) Fig. 4, 7 and 8 - professional scale bar markers should be provided.
- Figures edited.
7) SEM micrographs are provided for a subset of experiments. Could you please enclose justification why exactly these samples were visualized?
- The Figure 4 exposes the geometry of the particles in flight.
- The Figure 7 are examples of the characteristics/condition of the surfaces analyzed by EDS. The exemption of imperfections on the wire condition, the criteria of selection of particles collected during the flight (the analysis was on polished surfaces or transversal section and not on the free surface of the particles). The characteristics of the singles splats and the criteria of selection of the areas of these splats, looking the central part and not the splashes or arms.
- The Figure 8 reports the oxides formed during the melting of 309L on heat source, mainly because these oxides on the transversal section of the solidified material were not present on the condition of wire before the spraying.
8) Could you please comment on the reproducibility of this method? How many experiments were conducted for each parameter set?
- The measure of spray particles was done 3 times for each distance, material. After these measurements a more complete test was realized. The results presented same behavior of velocity, temperature, and diameter of particles, the collect and record of 5,000 particles was done for each material and distance to obtain the normal distributions of velocity, temperature and diameter of particles presented in this article.
Reviewer 3 Report
Article analysis particle properties of different steels. The article displays some nice results, however, I have few remarks:
1) Please increase the scaling of Figs. 3,5,6 and 7. The numbers on axis can be barely seen.
2) Please add scale bar in upper rights corner of the SEM images in Figs. 4 and 8.
3) Please proofread your article because there are some writing mistakes.
4) I missed more information of what has been already achieved by other authors in this field. Please add some numerical values or comparisons in order to highlight the novelty and better achievement in this field.
Author Response
1) Please increase the scaling of Figs. 3, 5, 6 and 7. The numbers on axis can be barely seen.
- Thanks for the criticism of the reviewer, the text was reviewed.
2) Please add scale bar in upper rights corner of the SEM images in Figs. 4 and 8.
- Figures edited and reviewed.
3) Please proofread your article because there are some writing mistakes.
- Text reviewed and new added authors edited. The authors believe the text quality was improved.
4) I missed more information of what has been already achieved by other authors in this field. Please add some numerical values or comparisons in order to highlight the novelty and better achievement in this field.
- Reviewed the introduction and the interpretation of results, improved the background and cited examples with numerical values, comparing the results obtained in experiments.
- The authors rewrite some parts of the text with intend to clarify the relevance of the text, for example on the abstract the sentence “The main goal of this research is evaluate how the velocity, temperature, and particle size are modified by the chemical composition of different materials (carbon steel, stainless steels, and FeMnCrSiNi alloy). The intention is predict how the modification of the process parameters will change the particles properties.” was included. The authors hope that this help to understand the relevance of this paper for the scientific and industrial community.
Reviewer 4 Report
Although the topic discussed in the present work is interesting, the literature review in the introduction were not sufficiently presented to point out the importance of this study. In addition, the clarity of figures (3, 5, and 6) should be improved. The figure's caption should be more informative. Therefore, I think that a minor revision should be given before acceptance this article in Coatings
Author Response
- Thanks to the reviewer, the introduction text was improved with intend to explain better the importance of the paper.
- Figures were reviewed.
- This study have a contribution to the scientific community and to the industry, since the materials evaluated (steels) are commonly applied and the results obtained help the users of ASP to confirm the parameter already in use, mainly the standoff distance, or change to improve the quality of the coatings.
- The alloy FeMnCrNi is new, under patent, and this evaluation of particles in-flight was never done before, configuring a novelty. The understand of its particles characteristics and behavior will support future studies of this alloy.
Round 2
Reviewer 1 Report
1 Five locations are listed in the experimental part, while only three results are showned in Figure 4.
2 There should be more SEM pictures for the collected particles.
3 EDS is not a quantitative methods for measuring compostion.
Author Response
1) Five locations are listed in the experiment part, while only three results are shown in Figure 4.
- Thanks for the consideration. The other histograms had been left out for the question of space and organization of the article. The authors believe all the histograms should be inserted as Supplementary Data. The file is attached.
2) There should be more SEM pictures for the collected particles.
- Thanks for the comment. There were inserted more pictures in Figure 2.
3) EDS is not a quantitative method for measuring composition.
- Thanks for the comment on this characteristic of the technique. The purpose of the evaluation was to compare the chemical composition in different conditions (feedstock wire, particles in flight, and splats solidified), and in this situation, the EDS is appropriate, since the qualitative comparison is achieved.

Reviewer 2 Report
Thank you for following my suggestions. I can now recommend publication of the article in the present form.
Author Response
The authors are glad to have met the expectations of the reviewer.
Reviewer 3 Report
All remarks are corrected.
Author Response

(The authors gave the same response as above.)
